# Exploring the Colors of Copper-Containing Pigments, Copper (II) Oxide and Malachite, and Their Origins in Ceramic Glazes

Iris Peng [1] , Katie Hills-Kimball [1], Isabela Miñana Lovelace [1], Junyu Wang [1], Matthew Rios [2], Ou Chen [1] and Li-Qiong Wang [1,*]

1   Department of Chemistry, Brown University, Providence, RI 02912, USA
2   Rhode Island School of Design, Providence, RI 02912, USA
*   Correspondence: li_qiong_wang@brown.edu

**Abstract:** The colors of copper-containing pigments, copper (II) oxide and malachite, and their origins in ceramic glazes were systematically examined over a wide firing temperature range using a suite of analytical and spectroscopy techniques including SEM, UV-Vis FORS, XRD, FTIR, and EPR to gain new insight into the structural and chemical transformations of the glaze during firing. The two colorants investigated were black copper (II) oxide (CuO) nanopowder and blue-green basic copper carbonate, or malachite ($Cu_2CO_3(OH)_2$), both of which produce a final light blue color following firing. Additionally, silicon carbide (SiC) was used to locally reduce CuO to simulate firing glazes in a reductive environment and produce a final red color. At lower temperatures, malachite was found to decompose to form CuO at 550 °C, elucidating the reason that two different copper colorants could be used interchangeably to form the same "Robin's Egg Blue" color. At 850 °C, a glaze sintering process occurred, resulting in the distribution of $Cu^{2+}$ in a square planar geometry and an observed blue color. This structural change occurred at temperatures lower than the glaze's melting point, indicating that complete vitrification of the glaze is not required for glaze coloration. Conversely, the reduction in $Cu^{2+}$ to $Cu^+$ through the addition of SiC did not occur until the glaze was fired above the melting temperature (1000 °C), signifying that high temperatures are required for the redox reaction to occur. This study sheds light on intermediate colorant-glaze interactions that are beneficial for understanding and predicting glaze coloring upon exposure to varying temperatures, and the results from this study can be applied to better-controlled glaze production for artists and a deeper appreciation of ceramic glaze chemistry and aesthetics.

**Keywords:** ceramic glazes; malachite; copper oxide; coordination chemistry; temperature; EPR



## 1. Introduction

Ceramic glazes, or glassy layers on top of fired clay bodies, were first produced around 1500 BCE in the ancient Near East [1]. Glaze production became desired because it strengthened fired clay pieces and rendered them nonporous, which made glazed containers ideal for storing liquids, especially for extended periods [1]. Glazed items also became valued for beautiful decorations and the vibrant colors they could impart on ceramic objects. Since the beginning of glaze production, the brilliant colors that have been achieved in pottery have been mainly due to the addition of metal oxides into the glaze [1–6]. For example, Egyptian faience, a self-glazing ceramic material that became popular around 2001–3000 BCE, was most commonly blue from the addition of copper, perhaps in the form of corroded copper metal or copper-containing minerals such as malachite or azurite [1]. However, the underlying mechanisms for the final glaze color's development throughout the firing process are not well studied.

Generally, glazes consist of a glass former, fluxes, refractories, and colorants [2,3]. The glass former, or silica ($SiO_2$), has a melting temperature that is too high for ceramic kilns, so fluxes are added to lower the melting temperature via weak intermolecular interactions [2].

Refractories, usually alumina ($Al_2O_3$), are added to prevent the molten glaze from running too much during the firing process [2]. Finally, glaze appearance can be customized with the addition of colorants, which are mostly transition metal oxides [4–6]. Even though colorants can impart vibrant colors to the final glaze products, they usually only make up a very small fraction of the glaze mixture (e.g., 1%) [3]. Additionally, their initial colors can be quite different from the final glaze colors [2,4].

The end color of the glaze is also affected by firing conditions and may vary based on whether the firing is performed in an oxidative or reductive kiln [4]. Since there are so many variables when creating glazed objects, artists often develop glazes empirically and through trial and error [5]. Thus, a deeper understanding of both the physical and chemical changes that the transition metal oxides in the glaze undergo during firing at a molecular level is needed to better understand, predict, and design final glaze colorings.

Copper-based transition metal oxides are commonly used as colorants in glazes to produce ceramic products with many different colors [1,3,4,6]. Specifically, copper colorants create blue and green colors in oxidative firing conditions due to a 2+ oxidation state, and they create red colors in reductive firing conditions due to an oxidation state and/or $Cu^0$ nanostructures [7–11].

One relatively simple and commonly used copper-containing glaze, "Robin's Egg Blue", is known to make a final blue color, but the exact mechanism of coloration for this and similar glazes throughout the entire firing process is not well understood. Furthermore, artists' glaze recipes may call for either copper oxide or copper carbonate as the colorant. Although there have been proposed mechanisms to explain the final blue color, the mechanisms are based on analyses of only the end products of the firing process [5,7,11–15]. A previous study on blue and green ancient Chinese faience found that the $Cu^{2+}$ ions are octahedrally coordinated [15]. Studies such as the one by Schabbach et al. [14]. have previously investigated different physical characteristics of glaze when fired at different final temperatures, but not throughout the entire firing process [14]. It is important to study the entire firing process since it enables the examination of the intermediate colorant-glaze interactions that are beneficial for understanding and predicting glaze coloring upon exposure to various temperatures. Thus, a more detailed, systematic study of the colorant and the chemical origin of the glaze color throughout the entire firing process is needed.

Here, we investigated the effect of firing temperature on glazes colored with copper (II) oxide (CuO) and malachite ($Cu_2CO_3(OH)_2$) transition metal oxides used in the "Robin's Egg Blue" glaze. Besides preparing blue glazes with each colorant, we also added silicon carbide (SiC) to a glaze with CuO to help create copper reds in an oxidizing atmosphere instead of the normal reduction kiln [4,16–18]. We studied glaze color as a function of increasing firing temperature systematically. Glaze samples taken at different temperature points throughout the firing process were analyzed to better understand the mechanisms behind the production of the final copper glaze colors. Unlike previous studies [1–6], a combination of several advanced characterization techniques, such as electron paramagnetic resonance (EPR) and fiber optic reflectance spectrophotometer (FORS), were utilized in this study to probe both the structural and chemical transformations of the glaze during firing at a wide range of temperatures. These results provide a better understanding of the colors of copper-containing pigments and their origins in ceramic glazes at the molecular level.

## 2. Materials and Methods

### 2.1. Bisque and Glaze Preparation

The Robin's Egg Blue glaze recipe and firing protocol (Table 1) were provided to us by Assistant Professor David Katz at the Rhode Island School of Design (RISD). Clay (gray, high-fire, Kentucky-Tennessee Clay), gerstley borate (U.S. Borax), nepheline syenite (Unimin), flint/silica (U.S. Silica), Edgar Plastic Kaolin (Edgar Minerals), and basic copper carbonate (malachite) (98–100%, World Metal) were obtained from the RISD Ceramics department's raw materials laboratory. We also obtained copper (II) oxide (CuO) nanopowder

(99%, 40 nm, U.S. Research Nanomaterials) and silicon carbide (SiC) powder (fine, 320 grit, Alfa Aesar).

**Table 1.** Bisque firing protocol.

| Temperature ($^\circ$C) | Time Spent at Temperature (min) |
|---|---|
| 66 | 60 |
| 93 | 60 |
| 121 | 180 |
| 1060 | 0 |

Clay was rolled into a small ball, flattened into a disk shape, and then left to dry in an ambient atmosphere. The disk was then fired in a ThermoFisher Lindberg/Blue 1200$^\circ$ LGO Laboratory Box Furnace to create the ceramic body (bisque) for us to glaze in a later step, according to the firing protocol (Table 1).

Once the furnace reached 1060 $^\circ$C, it was immediately turned off, and the disk was left to cool to room temperature overnight.

A base glaze without any colorant (based on a cone 08-06 "Robin's egg blue" glaze, melting temperature of approximately 1000 $^\circ$C) was created in four 5 g batches made up of 59.3% gerstley borate, 31.1% nepheline syenite, 2% Edgar Plastic Kaolin (EPK), and 7.6% flint/silica by mass. The percentages of individual compounds in the base glaze, such as $SiO_2$, $Al_2O_3$, $Fe_2O_3$, and CaO, were also calculated (Table S1). The powders were mixed with a mortar, pestle, and glass stirring rod. Each batch was then poured into a 15 mL scintillation vial using a funnel. An additional 1% (approx. 0.05 g) by mass of malachite was added to one vial, 1% CuO was added to a second vial, and 1% CuO and 1% SiC were added to a third vial. No colorant was added to the fourth vial. Approximately 8 mL of water was added to all four vials, and the glazes were mixed with a stirring rod and lightly agitated until the raw glaze resembled the consistency of heavy cream. The raw glazes were sealed with Parafilm until use.

### 2.2. Glaze Firing and Sample Collection

The four glazes were applied in spots onto the previously prepared bisque disk. They were also generously dotted onto a stainless-steel plate, allowing for easier sample removal. The 1% malachite, 1% CuO, and 1% CuO/1% SiC glazes were also painted onto 3 Coors 10 mL porcelain crucible lids per glaze type for better visualization of the glazes against a white background. All objects were fired in the Lindberg/Blue 1200$^\circ$ LGO Laboratory Box Furnace to 250 $^\circ$C, 550 $^\circ$C, 850 $^\circ$C, and 1000 $^\circ$C. Each temperature was held for 10 min before the objects were removed for visual observation and sample collection. For the glazed bisque disk and stainless-steel plate, after reaching 1000 $^\circ$C, the furnace was shut off, and the objects were left inside the furnace to slowly cool to room temperature overnight. For the glazed crucible lids, the 1% CuO and 1% malachite glazes were immediately removed from the furnace after reaching 1000 $^\circ$C and allowed to cool in an ambient environment. The 1% CuO/1% SiC glazed crucible lid was left in the furnace at 1000 $^\circ$C for an additional 20 min before being fired at 1050 $^\circ$C for 10 min. Afterward, the furnace was shut off, and the lid was left overnight in the furnace to allow the sample slowly cool to room temperature. Finally, the glaze samples on the stainless-steel plate were scraped off after cooling down to room temperature, stored in 15 mL scintillation vials, and sealed with Parafilm.

### 2.3. Characterization

All glaze samples fired at different temperatures were examined using a suite of analytical and spectroscopy techniques to characterize their physical and chemical properties.

2.3.1. Optical Properties

We observed the colors of our ceramic glazes as they went from their raw form to their final state during firing. The colors of the collected glaze samples, final glazes on the bisque

disk, final glazes on the crucible lids, and the colorants were examined by the naked eye and analyzed using UV-Vis reflectance spectroscopy. UV-Vis reflectance spectra were taken using an Ocean Optics UV-Vis fiber optic reflectance spectrophotometer (FORS), during which loose glaze samples were placed in between standard glass microscope slides that were taped together.

### 2.3.2. Physical and Chemical Properties

To observe the differences in glaze particle morphology with increasing firing temperature, an LEO-1530 VP scanning electron microscope (operating voltage 10 kV) was used for taking scanning electron microscopy (SEM) images of the 550 °C and 850 °C 1% malachite glaze samples. For this method, loose samples were pressed onto carbon backing before being attached to the sample holder and coated with 0.4 nm gold/platinum. We also analyzed the degree of crystallinity in all the glaze samples using X-ray diffraction (XRD), for which spectra were taken on a Bruker D8 Discover 2D High-Resolution X-ray diffractometer equipped with a Vantec 500 2D area detector operating with a Cu K-alpha ($\lambda = 1.541$ Å) radiation. Electron paramagnetic resonance (EPR) spectra were taken on a Bruker EMX Premium-X EPR spectrometer operating at room temperature with a 9.86 GHz frequency (X band), 2 mW power, and modulation amplitude of 1 G. Samples were loaded in 100 μL borosilicate glass capillaries and sealed with capillary sealing clay before being placed in quartz EPR sample tubes (4 mm thin wall, 250 mm length). The glass capillaries were also run without samples inside as control and subtracted from the sample spectra. Fourier-transform infrared (FTIR) spectra for glaze and colorant samples were taken using a Jasco 4100 FTIR spectrometer with an ATR attachment.

### 3. Results

#### 3.1. Optical Properties

Figure 1 illustrates that before firing, the no-colorant glaze was pale tan, while the 1% malachite glaze was light beige. Both the 1% CuO and 1% CuO/1% SiC glazes were gray, with the 1% CuO/1% SiC glaze being slightly darker due to SiC also being gray in color. After being fired to 250 °C, all the glazes were matte, dry, and powdery; besides being slightly lightened due to the evaporation of water, there were no significant color changes in the glazes. By 550 °C, the 1% malachite glaze darkened to a medium gray. There were no other significant color changes, and the glazes were still matte and powdery in texture but slightly more clumped together. After 850 °C, the no-colorant glaze was opaque white, while the other three glazes were all light blue, though the 1% CuO/1% SiC glaze appeared to have gray particles suspended inside. All the glazes were harder and more fused together, with a slight sheen. Above 1000 °C, on the bisque disk, the no-colorant glaze was transparent and colorless. The 1% malachite and 1% CuO glazes were both transparent, light blue, and very similar in color. The 1% CuO/1% SiC glaze was a light greenish-blue but appeared to include reddish areas (which can be better observed in the bottom image of Figure 2b). While the no-colorant, 1% malachite, and 1% CuO glazes were all smooth and shiny, the 1% CuO/1% SiC glaze had small bubbles on the surface, which made its appearance more opaque. For the glazed crucible lids (Figure 2a), the 1% CuO (Figure 2(a6)) and 1% malachite glazes (Figure 2(a7)) were a pale green when they were first removed from the furnace and into the ambient room conditions. They immediately cooled to solid glass and turned light blue (Figure 2b top and middle). Due to the rapid cooling rate, the two glazes underwent crazing, or spider web-patterned cracking due to fast-shrinking around the ceramic body, but were still highly reflective. After the 1%, CuO/1% SiC glaze in the crucible lid was cooled overnight, it had a light greenish-blue with small bubbles on the surface and red dots, which were significantly bigger and more spread out (Figure 2b bottom).

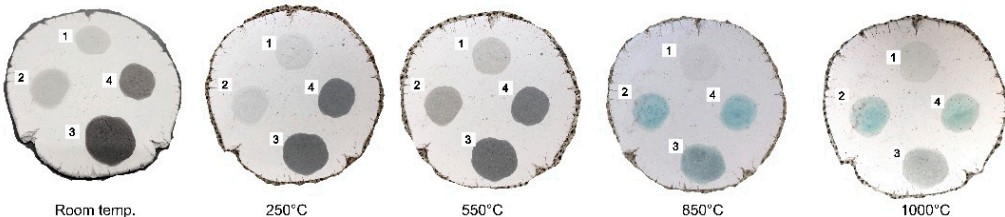

**Figure 1.** (**Left** to **right**): glazed bisque disk before firing, after 250 °C, after 550 °C, after 850 °C, and after 1000 °C. The glaze dots are labeled as (**1**) no colorant, (**2**) 1% malachite, (**3**) 1% CuO/1% SiC, and (**4**) 1% CuO.

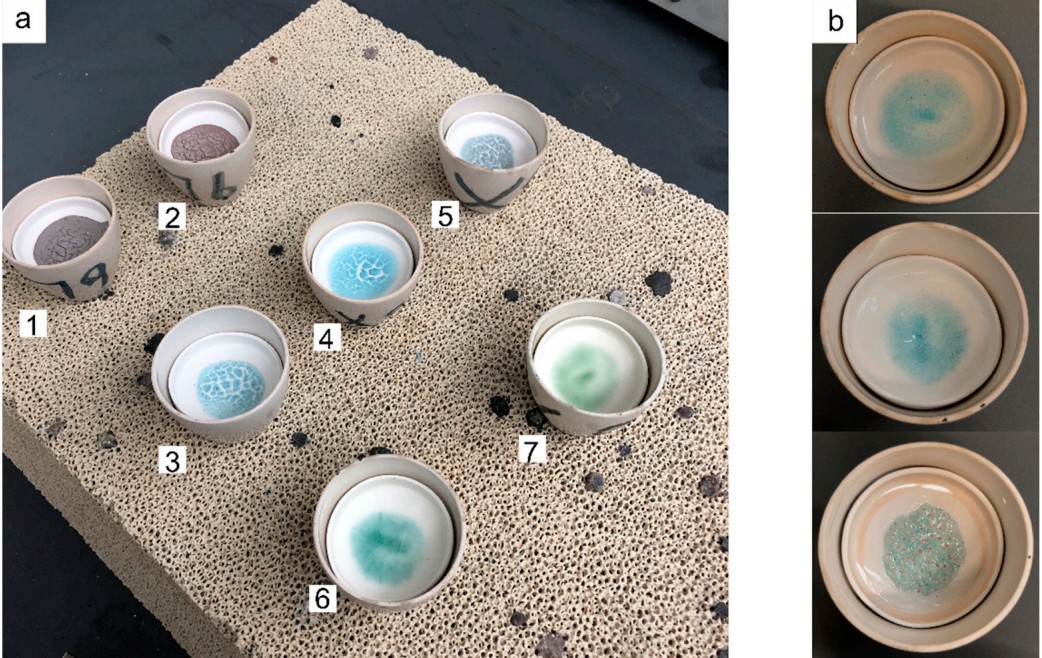

**Figure 2.** (**a**) Glazed crucible lids were prepared for 1% CuO (**1**) and 1% malachite (**2**) fired to 550 °C; 1% CuO (**3**), 1% malachite (**4**), and 1%CuO/1%SiC (**5**) fired to 850 °C, and the 1% CuO (**6**) and 1% malachite (**7**) glazes fired to 1000 °C directly taken out of the furnace, showing the initial green color. (**b**) Top to bottom: 1% malachite, 1% CuO, and 1% CuO/1% SiC glazes fired to 1000 °C in crucible lids following cooling.

FORS was used to measure the sample reflectance, illustrating how the glaze colors change throughout the firing process. As firing temperature increased, the reflectance peak shifted to bluer wavelengths for all of the colored glazes (Figure 3a,b), with peaks around the shorter wavelengths of visible light (i.e., ~500 nm). For the 1% CuO/1% SiC glazed crucible lid, when the probe was focused on a red dot, there was a reflectance peak at longer wavelengths (~650 nm) in addition to the blue peak, suggesting the co-existence of red and blue copper centers (Figure 3c, green line).

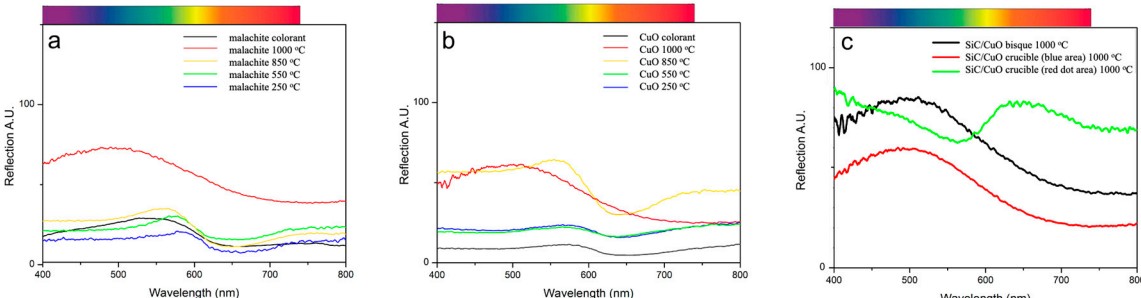

**Figure 3.** (**a**) Reflectance spectra for commercial malachite and the 1% malachite glaze series. (**b**) FORS reflectance spectra for CuO nanopowder and the 1% CuO glaze series. (**c**) Reflectance spectra for final 1% CuO/1% SiC glazes on different objects and in different areas.

### 3.2. Morphological and Structural Analyses

SEM imaging confirms that by 850 °C, the glazes have already begun the vitrification process or the transformation of a glaze made of a mixture of various crystalline particles into a non-crystalline glass. In the 850 °C sample, the particles were less defined and appeared to have begun sintering together (Figure 4b) compared to the more defined crystalline particles in a glaze mixture at 550 °C (Figure 4a).

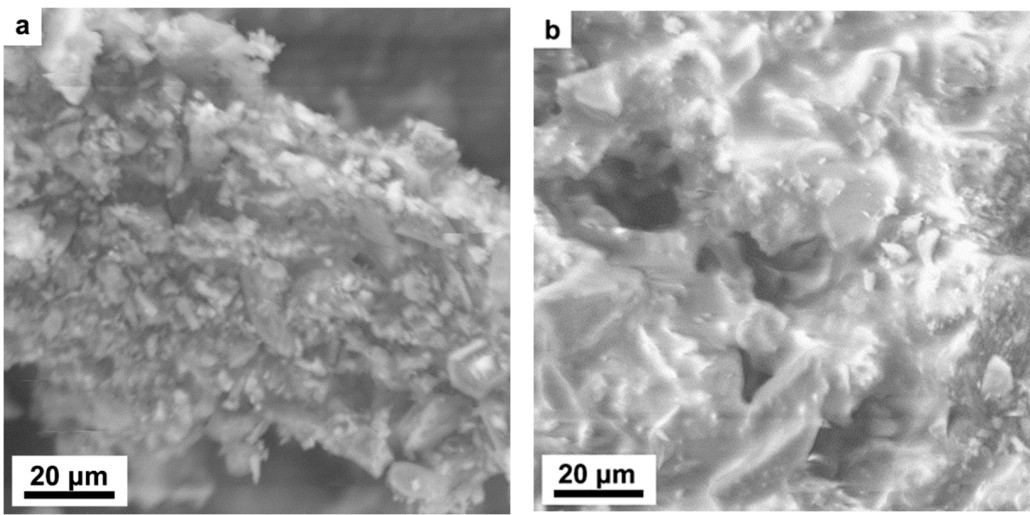

**Figure 4.** (**a**) SEM image for the 1% CuO glaze fired to 550 °C compared to the (**b**) 1% CuO glaze fired to 850 °C.

XRD was used to monitor changes in crystallinity in a glaze during the increase in firing temperature. As expected, as firing temperature increased to the flux point (~1000 °C) of all of the glazes, they changed from a physical mixture of multiple crystalline materials into an amorphous, aluminosilicate glass, as indicated by the XRD pattern of the 1% CuO sample fired to increasing temperatures (Figure S1). Specifically, all diffraction peaks became broader upon heating to 850 °C, supporting the sintering observed in the SEM measurements, and a fully amorphous structure was reached at 1000 °C. However, because the colorants (CuO or malachite) only made up 1% wt of the glazes, their XRD signatures were not discernable among the stronger signals of the other glaze ingredients.

EPR was employed to further investigate the structural changes around the transition metal center ions in the no-colorant, 1% malachite, and 1% CuO glaze samples fired to 850 °C (Figure 5a), respectively. The EPR spectrum of the glaze without colorant (Figure 5a, blue) possessed two signals from the presence of $Fe^{3+}$ in the glaze components. Specifically, the sharper peak with g ~4.3 is attributed to isolated high spin $Fe^{3+}$ (S = 5/2) ions in a distorted tetrahedral or octahedral environment [19–21]. The weaker broad signal

centered with a g value of ~2.0 is likely due to the superexchange coupling of $Fe^{3+}$ centers and/or surface defects [19–21]. In addition to the same $Fe^{3+}$ features, the samples with 1% malachite (Figure 5a,b, red) and 1% CuO (Figure 5a,b, black) possessed an anisotropic peak of ~ 3450 G as a result of the presence of $Cu^{2+}$. EPR was also taken for the 1% malachite glaze at other firing temperatures below 1000 °C (Figure 5c). For the glaze samples fired to 250 °C (Figure 5c, gray) and 550 °C (Figure 5c, dark red), the same $Fe^{3+}$ peaks were observed at g values of ~4.2 and ~2.0, indicating that the coordination of $Fe^{3+}$ remains constant throughout the entire temperature range studied. In addition, we observed a sharp signal with six hyperfine splitting and additional satellite peaks, which could possibly be attributed to trace amounts of $Mn^{2+}$ present in the glaze components, as seen in previous studies [22]. This signal disappeared following heating to 850 °C, possibly as a result of the oxidation of $Mn^{2+}$ to $Mn^{3+}$ (Figure 5c, red), an ion that EPR can no longer detect [22].

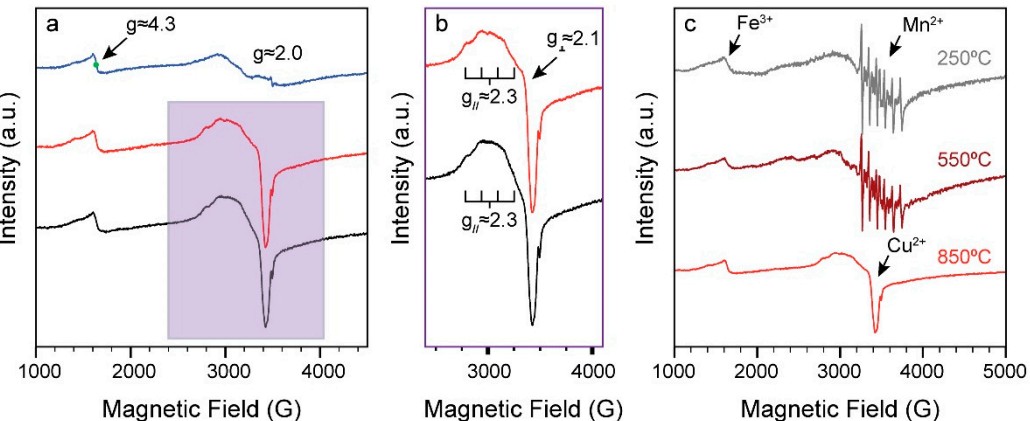

**Figure 5.** (**a**) EPR spectra for the no colorant (blue), 1% malachite (red), and 1% CuO (black) glazes fired to 850 °C. (**b**) Zoomed in EPR spectra for the 1% malachite (red) and 1% CuO (black) shown in the purple highlighted box in (**a**). (**c**) Normalized EPR spectra for the 1% malachite glaze series below 1000 °C (gray, dark red, and red fired to 250 °C, 550 °C, and 850 °C, respectively).

As expected, FTIR analysis of the glaze samples, such as in the 1% CuO glaze (Figure S2a), illustrated the complete loss of both free and crystal-bound water by 550 °C, as indicated by the loss of the broad peak around 3300 $cm^{-1}$ with increasing temperature (Figure S2b).

## 4. Discussion

The color of the final glaze product depends on many factors, such as the type of transition metal oxides used as colorants and their oxidation states after firing. The origin of the blue color in the 1% CuO and 1% malachite glaze samples is attributed to the d-d electronic transition involving unfilled d orbitals and the $Cu^{2+}$ coordinated complexes. Transition metal ions such as $Cu^{2+}$ possess unfilled d orbitals, which degenerate when no ligands are present [16,23,24]. However, when ligands surround and interact with a central transition metal ion, the ligand can donate electrons into the metal ion's unfilled d orbitals (or vice versa). This causes the metal ion's d orbitals to split and have different energy levels, resulting in the ligand field effect. Based on the energy gap size, electrons in the lower-energy d orbitals can absorb certain wavelengths in the visible light spectrum and become excited to a higher-level d orbital. It is this absorption of visible light that causes many transition metal complexes to appear colored. The color of the complex depends on the number of d-electrons, the type of coordination ligands around the metal ion, and the geometry of the ligand arrangement. The blue color in the 1% CuO and 1% malachite glaze samples is due to the absorption of the red light with a wavelength between 610 and 650 nm and the subsequent reflection of its complementary blue-green color. However, the origin of the red color in the final glaze (see Figures 1 and 2) with SiC as a reducing agent during firing is different from that of the blue color under the oxidative condition.

SiC has been used to reduce $Cu^{2+}$ to $Cu^{+1}$ or $Cu^0$ [17]. The red color is certainly not from the d-d transitions because of no unfilled d orbitals in $Cu^{+1}$ or $Cu^0$. Thus the red color is either from the dispersed red-colored $Cu_2O$ ($Cu^{+1}$) particles in the final glaze or nanosized metallic copper ($Cu^0$) [9,17].

It is clear from our combined results that the colors of copper-containing glazes are not due to the colorant in the physical mixture of the raw glaze but rather to a change in overall glaze structure as the firing temperature approaches and exceeds 850 °C. For example, the color of CuO is black in the raw glaze but blue color in the final glaze product. Our data, especially from the EPR measurements, further confirm that the origin of the final blue colors in the glaze under oxidative conditions is due to the transition metal $Cu^{2+}$ ions that are coordinated with the surrounding glass matrix since EPR spectra are sensitive to any changes in either the oxidation state of the central transition metal ions or the ligands that are coordinated to the center ions. EPR data showed that copper in the 1% malachite and 1% CuO glazes had an oxidation state of $Cu^{2+}$ throughout the firing process, even when the glazes were not yet blue, indicating that no redox reactions occurred (Figure 5). Specifically, a broad peak centered at ~3370 G observed in the 1% malachite glaze at both 250 °C and 550 °C is due to the presence of coupled $Cu^{2+}$ centers, in agreement with the previous study [25], while an anisotropic peak observed at 850 °C is attributed to individual $Cu^{2+}$ ions within a different structural environment (Figure 5c). EPR data further confirm that the new coordinated $Cu^{2+}$ complexes are formed after heating the glaze to 850 °C due to the interaction with the glass matrix under high temperatures.

Interestingly, we observed that the 1% malachite glaze changed from beige to darker gray, more similar to the 1% CuO glaze, as the firing temperature reached 550 °C (Figure 1). Natural malachite is known to decompose into CuO at about 375 °C, for which Frost et al. (2002) proposed the decomposition mechanism to be as follows [26]:

$$Cu_2CO_3(OH)_2 \rightarrow 2CuO + H_2O + CO_2$$

This observation suggests that this decomposition is also occurring within the glaze mixture as well, specifically before the glazes reach a high enough temperature to begin sintering. This decomposition also explains why the final colors of the 1% CuO glaze and 1% malachite glazes were very similar, as shown by the similarities in the FORS spectra of the glazes fired to 1000 °C (Figure 3a,b). These results indicated that both CuO and malachite can be used as the colorant to create "Robin's egg blue." Furthermore, the observation of similar anisotropic $Cu^{2+}$ EPR lines (Figure 5b) for both the 1% malachite and 1% CuO glazes upon heating to 850 °C proves that both colorants give an identical $Cu^{2+}$ coordination environment in the final glaze. Thus the origin of the blue colors in both 1% malachite and 1% CuO glaze heated to high temperatures is due to the similar coordinated $Cu^{2+}$ complex with the surrounding glass matrix.

The change in glaze color from gray to blue by 850 °C for all of the colored glazes was a surprising observation, as 850 °C was below the flux point of the glaze (Figure 1). It suggests that even before the glaze became completely molten, $Cu^{2+}$ was already well-distributed and coordinated with the other glaze compounds, likely in an octahedral-like and symmetrical structure due to the light blue color [11]. This was further evidenced by the EPR spectra for samples fired at 850 °C, where an anisotropic $Cu^{2+}$ signal strongly appeared at ~3450 G (Figure 5). Specifically, the fact that $g_\parallel > g_\perp > 2.0$ confirmed that $Cu^{2+}$ is located in a square planner ($D_{4h}$) geometry due to the Jahn Teller distortion [21,25,27,28]. Four hyperfine splitting signal as a result of the interaction between the unpaired electrons and the nuclei ($I = 3/2$) was partially resolved for the parallel component ($g_\parallel \approx 2.3$, $A_\parallel \approx 135$ G) but could not be resolved for the perpendicular component ($g_\perp \approx 2.1$) (Figure 5b). Overall, our signal matches well with other examples of $Cu^{2+}$ in square planar coordination [21,25,27–29]. Additionally, it is probable that a second broad $Cu^{2+}$ peak is contributing to the overall signal as a result of some $Cu^{2+}$ ions that remain strongly coupled in a manner like $Cu^{2+}$ in the raw glaze, similar to previous reports [21,28]. Taken together, the significant color change and dramatic sharpening of the EPR signal, even before the glass was molten,

validate that the sintering process for this glaze already begins at 850 °C. This can be further seen in the SEM image (Figure 4) comparisons and in changes in XRD (Figure S1).

Interestingly, the glazed crucible lids became green immediately after they were taken out of the furnace (Figure 2a). This may be due to the $Cu^{2+}$ being in a more tetrahedral environment with the superheated glaze matrix since aqueous $CuCl_4$ is known to be green [30]. Alternatively, the electronic transition or degree of axial elongation may have been different due to the effects of high-temperature heating, causing us to see the unexpected green color. The cooling process may have then changed the structure of the glaze matrix such that square planar geometry coordination was favored, leading to the blue color of the final glaze product at room temperature (Figures 1 and 2).

For the 1% CuO/1% SiC glaze, it was observed that the red spots did not appear until after the glazes were fired to 1000 °C and above (Figures 1 and 2b). At 850 °C, the glaze was light blue with gray particulates, which were most likely the unreacted SiC particles (Figure 1). The red dots were more significant when the glaze was held for a longer time and at a higher temperature than 1000 °C, as seen when comparing the glazed crucible lid to the glazed bisque disk (Figure 2b). These results suggest that longer soaking times, a.k.a. the time the glazes are held at the highest temperature, are needed for artists who hope to create full, copper red glazes in an oxidative environment using SiC as a reducing agent. Comparatively, because the final color for our copper blue glazes (no SiC) was achieved before total vitrification to glass, an artist's soaking time for copper blue glazes is only dependent on the desired final surface texture (i.e., smoother and glassy vs. rougher). A mechanism for the reduction in copper using SiC was previously proposed by Orna and Goodstein [16]:

$$8CuO + SiC \rightarrow 4Cu_2O + SiO_2 + CO_2$$

This proposed mechanism was supported by our observations of the reduction in red spots we observed in glaze with SiC, as $Cu_2O$ (Cu has a 1+ oxidation state) is known to be red [16]. Additionally, the non-red glaze areas appeared greener than the blue 1% CuO and 1% malachite glazes, suggesting that $Cu^+$ was also distributed along with the $Cu^{2+}$ in the non-visible red areas (Figures 2b and 3c) [31]. In addition, the final glaze was not as reflective as the other glazes without SiC due to the presence of small bubbles, which were most likely caused by $CO_2$ production during the copper reduction process. The high temperature that the reaction required to take place was likely influenced by the fact that SiC is a refractory material. Nevertheless, there is still disagreement in the literature as to whether copper reds are due to the presence of $Cu_2O$ or $Cu^0$ nanoparticles after reduction [7,9–11,17]. More careful analyses, such as XANES, would help to confirm the underlying chemical changes that result in these red areas.

Figure 6 is an illustration that summarizes the proposed mechanism and the timelines of the firing process for our "Robin's egg blue" glaze variants based on the previous literature and the data we collected in this study [16,23,26]. It illustrates the loss of water in all glazes mostly by 250 °C, as indicated by the size of the blue arrows, but water may have still been continuously lost with higher temperatures as other glaze components begin to decompose and sinter, as indicated by the increasingly smaller blue arrows. The water loss and loss of volume of the glaze were reflected in the gradually decreasing glaze square size and thickness as the furnace temperatures increased. By 850 °C, all glaze samples appeared blue in color (Figures 1 and 2), indicating that complexation with the $Cu^{2+}$ ions had occurred, which may have been due to the formation of silica-alumina matrix as the glaze components began to sinter. Finally, by 1000 °C and above, the glazes vitrified into amorphous glass, resulting in disorder in the silica-alumina matrix that was indicated by the loss of defined peaks in the XRD pattern (Figure S1). Specifically for the 1% malachite glaze, the malachite likely decomposed into CuO by 550 °C, as represented by the darkening of the turquoise-colored colorant dots into black dots along with a loss of $H_2O$ and $CO_2$. Meanwhile, for the 1% SiC/1% CuO glaze, the SiC remained unreacted until around 1000 °C, before it reduced CuO to likely form $Cu_2O$, $SiO_2$, and $CO_2$, represented by the red spots and yellow particles at the right bottom of Figure 6.

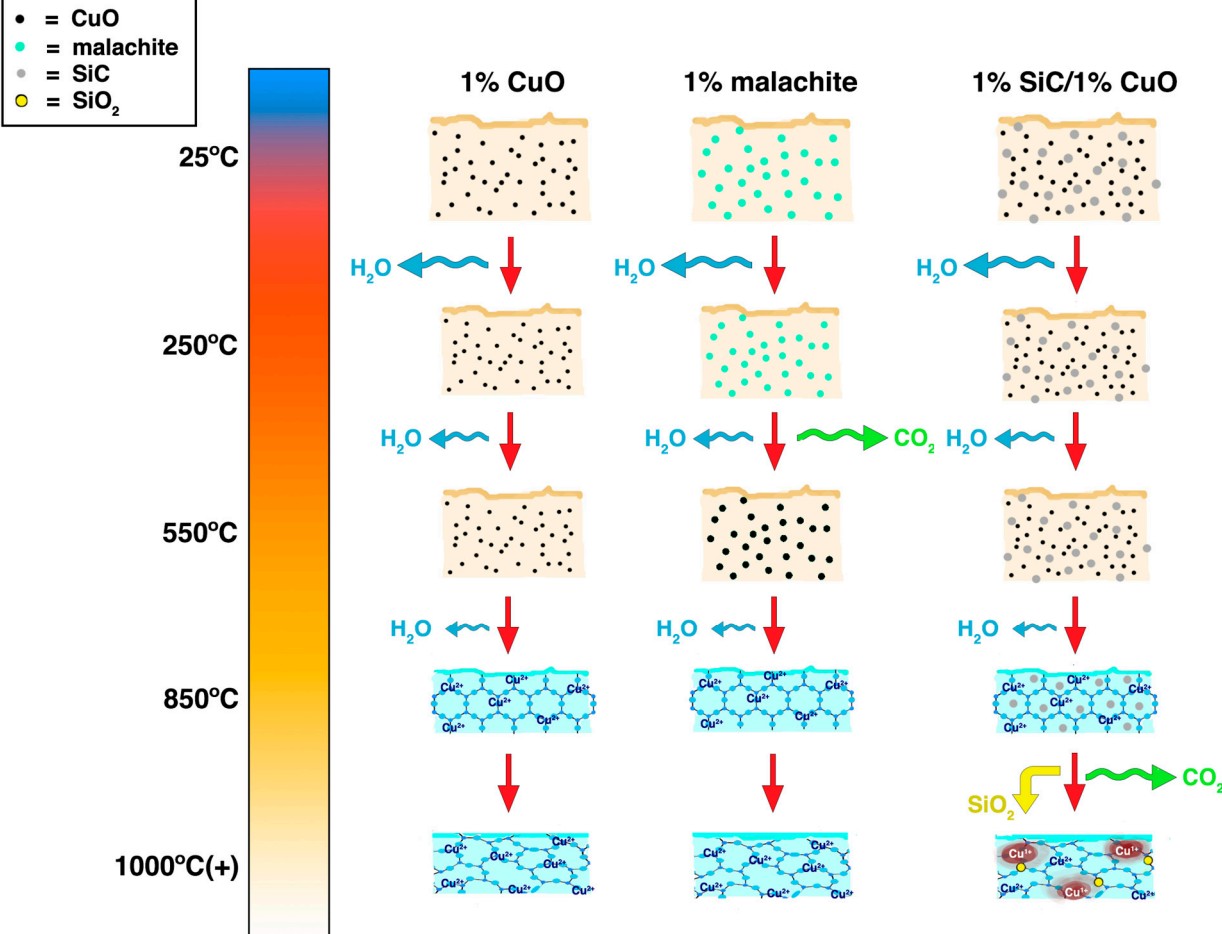

**Figure 6.** A schematic illustration showing the proposed mechanisms and timelines of the firing process for the "Robin's egg blue" glaze variations. By 850 °C, the presence of color due to $Cu^{2+}$ coordination suggests the presence of a silica-alumina matrix (represented by the regular ordered lattice structure), which becomes more disordered as the glaze becomes glass but ultimately still allows for coordination.

To further elucidate the chemistry of glaze firing, X-ray absorption near edge structure (XANES) could be performed to confirm the electronic transitions that are responsible for the color of our blue glazes. Other methods, such as X-ray photoelectron spectroscopy (XPS), can also be used to further investigate the form(s) of copper responsible for our copper red glaze.

This study includes a novel, systematic study of glaze coloration and chemistry as a function of firing temperature, particularly for an artistic ceramic glaze that only contains 1 wt. % copper colorant. We unexpectedly observed that final glaze colors were achieved below the glaze flux range, indicating that copper complexation likely occurred at the very beginning of the vitrification stage. Additionally, the decomposition of malachite into CuO may have been observed by the midpoint of the firing process, explaining why either CuO or malachite can be used by artists to achieve the same color.

Our results have provided a better understanding of how only 1 wt. % copper compound can produce vibrant glaze colors that are not due to the initial, intrinsic color of the colorant but rather an overall structural change in the glaze that causes the surrounding glaze matrix to interact with the copper centers. Additionally, we gain a better understanding of the chemical reactions behind the color changes and when the reactions take place during the firing process. These results can be applied to better-controlled glaze production for artists and a deeper appreciation of ceramic glaze chemistry and aesthetics.

## 5. Conclusions

The effect of firing temperature over a wide range on glazes colored with copper (II) oxide (CuO) and malachite ($Cu_2CO_3(OH)_2$)—transition metal oxides used in "Robin's Egg Blue"—has been studied systematically using a combination of techniques including SEM, UV-Vis reflectance spectroscopy, XRD, FTIR, and EPR. Glaze samples taken at different temperature points throughout the firing process were analyzed to better understand the mechanisms behind the production of the final copper glaze colors. First, at 550 °C, the malachite glaze darkened in color, pointing to the decomposition of malachite into CuO, suggesting the reason why two different copper transition metal complexes can be used interchangeably by ceramic artists. At 850 °C, a glaze sintering process had begun, resulting in the distribution of $Cu^{2+}$ in a square planar geometry and a light blue color. This structural change occurred at temperatures below the glaze's melting point, suggesting that complete vitrification of the glaze is not required for glaze coloration. In contrast, the reduction in $Cu^{2+}$ through the addition of SiC did not occur until the glaze was fired above the melting temperature (1000 °C), signifying that a relatively high temperature (>1000 °C) is required for this redox reaction to occur. Our study presented here sheds light on intermediate colorant-glaze interactions that are beneficial for understanding and predicting glaze coloring upon exposure to varying temperatures.

**Supplementary Materials:** The following supporting information can be downloaded at: https://www.mdpi.com/article/10.3390/colorants1040023/s1, Figure S1. XRD spectra for the 1% CuO glaze series; Figure S2. (a) FTIR spectra of 1% CuO glazes before firing, after 250 °C, after 550 °C, and after 850 °C. (b) Zoomed-in area of the red box on Figure S2a, illustrating the broad, water stretch peaks around 3300 cm$^{-1}$. The signal gradually approaches 100% transmission with increasing firing temperatures, indicating that both free and crystal-bound water is lost throughout the firing process; Table S1. Percentage of different compounds in the base glaze (no colorant) calculated based on values from Digitalfire, an online reference for ceramic artists. Generally, RO = flux, R2O3 = intermediate, RO2 = glass former *.

**Author Contributions:** Conceptualization, L.-Q.W. and I.P.; methodology, I.P., K.H.-K., I.M.L., J.W., O.C. and L.-Q.W.; validation, I.P., K.H.-K. and I.M.L.; formal analysis, I.P., K.H.-K., I.M.L. and J.W.; investigation, I.P., K.H.-K., I.M.L., J.W. and L.-Q.W.; resources, M.R.; writing—original draft preparation, I.P.; writing—review and editing, I.P., K.H.-K., I.M.L., J.W., M.R., O.C. and L.-Q.W.; visualization, I.P.; supervision, L.-Q.W.; project administration, L.-Q.W.; funding acquisition, L.-Q.W. All authors have read and agreed to the published version of the manuscript.

**Funding:** This research received no external funding.

**Institutional Review Board Statement:** Not applicable.

**Informed Consent Statement:** Not applicable.

**Data Availability Statement:** The data presented in this study are available in this article.

**Acknowledgments:** This research was funded and supported by the Karen T. Romer Undergraduate Teaching and Research Award Program at Brown University. We are grateful to Hector Garces, Anthony McCormick, and the Institute for Molecular and Nanoscale Innovation for their help with using EPR and SEM machinery, as well as David Katz (Department of Ceramics, Rhode Island School of Design) for his expertise, glaze recipe, and materials. Ou Chen would also like to thank the startup support from Brown University.

**Conflicts of Interest:** The authors declare no conflict of interest.

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
