# Peer review of "Exploring the Colors of Copper-Containing Pigments, Copper (II) Oxide and Malachite, and Their Origins in Ceramic Glazes"

_2079-6447, doi:10.3390/colorants1040023_

Round 1
Reviewer 1 Report
Dear authors,
the idea of contribution is interesting, however the manuscript need major revision in all chapters. The abstract attracts readers but the main text need to be improved.
Firstly the technical aspect of the form of the contribution is very low and need to be enhanced: the first table in chapter 2.1 did not have the label and have no link in text; Figure 1 and 2 have changed labels and the documentation could be more precise; table in supplement need to be add how the calculation was performed; Figure 3 a nad b are the same spectra; SEM images are very low in quality; tables of all figures are too complicated and could have better links to text; legends in figures could be improved to better interconnected individual results; first two paragraphs of discussion have to be moved to introduction.
Secondly the outcome of the contribution:
I deeply recommend supplement knowledge in the field of ionic dyes and the equilibrium state of copper ions in an amorphous matrix.
Introduction should be edited
- mainly the first reference is focused on Egyptian Blue and the introduction have to be further extended by this type of pigment because it is very important for the following experiment in terms of the historical context,
- an adequate introduction must be added to the use of copper-based ionic dyes,
- the use of ionic dyes is well studied but it is not known how this was done historically - this part 2nd paragraph page 2 must be improved in this context and also the reaction of ionic dyes,
- the first two paragraphs of discussion part have to be moved into introduction part.
Materials and Methods part is complicated and confusing:
- the preparation chapter (2.1) is vaguely and complicatedly described,
- the table in this part is without any connection to text and label is missing, the choice of firing regime must be commented,
- the final samples compostion were selected on basis of what - this theme should be improved.
The part of results is poor due to badly chosen experiment for selected methods:
- XRD and FTIR results have no benefit and absolutely lacking in meaning,
- optical properties are interesting but the figures could be improved, labels are tossed - Figure 1 and 2 have exchanged labels, and the connection of this part to further results and discussion part is missing,
- Figure 3 a and b are the same and the correct spectra should be added, Figure 3 c legend could be improved and the interpretation of the reflectance spectra should be extended by discussion not just a description of the results,
- the SEM results are only from morphology, but the microchemical analysis (EDS or WDS) will clearly improve the result parts, e.g. mapping of Cu distribution in the glaze, the focus of SEM images is of low quality,
- the most beneficial are EPR results but however need some additional description - the only method of structural changes description - the changes are not proven and the interpretation of EPR results should be connected to some references if the work is based on these results. The the presence of Mn2+ is not supported by any other method.
Discussion part is very repetitive and descriptive:
- the 1st and 2nd paragraphs are introduction as it was mentioned previously,
- the first sentence of 3rd paragraph should be more detaily commented,
- the glaze has no structure,
- the decomposition of malachite is familiar,
- the results must be supported by other analyzes or a part of the experiment must be modified to achieve the pivotal results.
- Figure 6 have very complicated and long description which is repeated in the discussion itself and the 3rd column is not fully supported by results and could only be reported hypothetically.
The experimental parameters could be improved to obtain key results which will support the suitability of the contribution.
Yours faithfully.
Author Response
General comments: The idea of contribution is interesting, however the manuscript need major revision in all chapters. The abstract attracts readers but the main text need to be improved.
Responses: We appreciate your valuable comments. We have made all necessary changes based on your feedback. In addition, we also carefully edited the entire manuscript to make sure our ideas and results are clearly presented. We believe our revised manuscript is much stronger. Please see the following point-by-point responses.
Comments: Firstly the technical aspect of the form of the contribution is very low and need to be enhanced: the first table in chapter 2.1 did not have the label and have no link in text; Figure 1 and 2 have changed labels and the documentation could be more precise; table in supplement need to be add how the calculation was performed; Figure 3 and b are the same spectra; SEM images are very low in quality; tables of all figures are too complicated and could have better links to text; legends in figures could be improved to better interconnected individual results; first two paragraphs of discussion have to be moved to introduction. I deeply recommend supplement knowledge in the field of ionic dyes and the equilibrium state of copper ions in an amorphous matrix.
Responses:
- As per your suggestions, we labeled the firing scheme table in section 2.1.
- The legends for Figures 1 and 2 were swapped due to formatting errors, and have been corrected.
- We added a sentence on how the calculation was performed to a table in the supplement.
- We replaced the spectrum duplication in Figure 3, and, for ease of interpretation, we have altered the colors in the key to be more consistent between the sub-figures.
- Thank you for pointing out that the SEM images in Fig. 4 are very low in quality. Due to the non-conductive nature of the Cu glazes, it is difficult to obtain high-quality images at a high resolution as shown in Fig. 4. In order to show the vitrification differences between glazes in firing temperatures, we decided to keep the higher magnification images (Fig. 4) since Fig. 4 clearly shows that by 850ºC, the glazes have already begun the vitrification process or the transformation of a glaze made of a mixture of various crystalline particles into a non-crystalline glass. In the 850ºC sample, the particles were less defined and appeared to have begun sintering together (Fig. 4b), compared to the more defined crystalline particles in a glaze mixture at 550ºC
- We simplified several figure captions and made a better tie to the text.
- We moved the first paragraph of the discussion into the introduction and rewrote the second paragraph to explain the origin of color in the final glaze product.
- We added a couple of paragraphs discussing the origin of the blue color in the final glaze, which is attributed to the Cu2+ ion complex in an amorphous matrix. However, the detailed information on the ionic dyes and the equilibrium state of copper ions in an amorphous matrix requires more advanced techniques such as EXAFS (extended x-ray absorption fine structure) with bright synchrotron radiation light source due to the low copper contents in the glaze mixtures.
Comments: Introduction should be edited
- mainly the first reference is focused on Egyptian Blue and the introduction have to be further extended by this type of pigment because it is very important for the following experiment in terms of the historical context,
- an adequate introduction must be added to the use of copper-based ionic dyes,
- the use of ionic dyes is well studied but it is not known how this was done historically - this part 2nd paragraph page 2 must be improved in this context and also the reaction of ionic dyes,
Responses: Thank you for your insightful suggestions. We have included more details about Egyptian blue and its coloration through copper from Tite et al. (reference 1) as suggested, including in the transition metal complexation paragraph which has been moved to page 3, as mentioned previously, to better introduce copper-containing colorants in ceramic glazes. One such addition to the introduction on this topic is as follows: “For example, Egyptian faience, a self-glazing ceramic material that became popular around 2001-3000 BCE, was most commonly blue from the addition of copper perhaps in the form of corroded copper metal or copper-containing minerals like malachite or azurite.”
We edited the introduction to address most of your concerns. We also discuss the ionic dye and the chemical origin of the color in the discussion section.
Comments: Materials and Methods part is complicated and confusing:
- the preparation chapter (2.1) is vaguely and complicatedly described,
- the table in this part is without any connection to text and label is missing, the choice of firing regime must be commented,
- the final samples composition were selected on basis of what - this theme should be improved.
Responses: In response to your helpful comments, we have simplified the methods section.
- We added an explanation as to why we chose this firing regime and glaze recipe with the following note: “The Robin’s Egg Blue recipe and firing protocol (Table 1) was provided to us by Assistant Professor David Katz at RISD.” The glaze recipe and protocol were provided to us by a ceramics professor at RISD, David Katz, for a cone 08-06 glaze. This metric is used by artists to describe a glaze that should be fired to a final temperature of about 1000℃ to become glassy (anywhere from 922-1013 degrees based on pyrometric cone type).
- A label has been added to Table 1 on page 6. We added a description for the choice of fire regime as follows: “The disk was then fired in a ThermoFisher Lindberg/Blue 1200º LGO Laboratory Box Furnace to create the ceramic body (bisque) for us to glaze in a later step, according to the firing protocol (Table 1).”
- The final sample composition is merely based on the commonly used “Robin’s Egg Blue” recipe” provided by our collaborator. The results from this study apply to the other types of glazes.
Comments: The part of results is poor due to badly chosen experiment for selected methods:
- XRD and FTIR results have no benefit and absolutely lacking in meaning,
- optical properties are interesting but the figures could be improved, labels are tossed - Figure 1 and 2 have exchanged labels and the connection of this part to further results and discussion part is missing,
- Figure 3 a and b are the same and the correct spectra should be added, Figure 3 c legend could be improved and the interpretation of the reflectance spectra should be extended by discussion not just a description of the results,
- the SEM results are only from morphology, but the microchemical analysis (EDS or WDS) will clearly improve the result parts, e.g. mapping of Cu distribution in the glaze, the focus of SEM images is of low quality,
- the most beneficial are EPR results but however need some additional description - the only method of structural changes description - the changes are not proven and the interpretation of EPR results should be connected to some references if the work is based on these results. The presence of Mn2+ is not supported by any other method.
Responses: In response to your feedback, we have made the following changes.
- One of the goals of this research was to take snapshots of the glaze’s optical, physical, and chemical properties throughout the firing process. While our XRD and FTIR results may be expected, they gave us insight into approximate temperatures during which changes happen in the glazes.
- The legends for Figures 1 and 2 were swapped due to formatting errors, and have been corrected.
- We have replaced the spectrum duplication in Figure 3, and, for ease of interpretation, we have altered the colors in the key to be more consistent between the sub-figures.
- While we conducted research with EDX, our results only showed the elemental makeup of the glaze, and the percentage of Cu maybe doesn’t change in a specific area even if the individual centers become complex with the surrounding glaze components. The presence of Mn2+, however, does show in the EDX data. We decided not to include EDX data since they did not provide essential information (we did not perform mapping of the Cu distribution in the glaze). While there are Mn peaks, it does not tell us the valence.
- In this study, EPR is most beneficial for probing the oxidation state and the surrounding chemical environment of copper ions. We have added more discussion on how EPR is used to get such information and related to the previous work.
Comments: Discussion part is very repetitive and descriptive:
- the 1st and 2nd paragraphs are introduction as it was mentioned previously,
- the first sentence of 3rd paragraph should be more detaily commented,
- the glaze has no structure,
- the decomposition of malachite is familiar,
- the results must be supported by other analyzes or a part of the experiment must be modified to achieve the pivotal results.
- Figure 6 have very complicated and long description which is repeated in the discussion itself and the 3rd column is not fully supported by results and could only be reported hypothetically.
Responses: We have trimmed the discussion section to improve the flow as per your helpful feedback and added necessary discussions on the chemical origin of the color and how EPR is a powerful tool to probe such information.
- We qualified the observation of our 1% malachite glaze becoming darker by 550℃ being caused by thermal decomposition into CuO, as we are aware of the known decomposition of natural malachite with increasing temperature, according to Frost et al. (2002).
- Unfortunately we did not conduct FTIR analysis on the 1% malachite glaze, which perhaps could have shown a decrease in the CO3 asymmetric stretching around 1500-1400 cm-1. Then again, since malachite is only 1% of the glaze recipe we could have also not been able to see that peak disappear with increasing temperatures.
- We shorten the description of Figure 6 and the 3rd column is partially supported by our results through comparative examination with other glazes and the red-colored dots due to the reduction of the copper ions. We added that the sentence “ We recognize that future analyses such as XANES would be useful in confirming the mechanisms of copper reduction from 2+ to 1+ or even to 0 (nanoparticles).”
Comments: The experimental parameters could be improved to obtain key results which will support the suitability of the contribution.
Responses: Thank you for this comment. We look forward to future analyses that can be used to further investigate the mechanisms of the glaze color development we have observed in our study. The main focus of this work is to study the changes in the glazes as a function of temperature which has not been studied previously. The results will help the artist to understand the color of the glazes as a function of the firing temperature. The detailed composition and structures of the Cu glaze require more study in the future involving more advanced techniques such as XANES.

Reviewer 2 Report
This paper by Peng & al is intersting, clear and well written. My few comments are:
- in the introduction, it is written than ceramic glazes were first produced roughly 10,000 years ago in the ancient Near East but they would rather appear around 1500 BC in the Near East (as mentioned in the reference)
- legends of Figure 1 and 2 are reversed. In the text, the refences to Fig 1 and 2 have to be checked
Author Response
General comment: This paper by Peng & al is interesting, clear and well written.
Response: We would like to thank the referee for the positive comments and the valuable suggestions. We have addressed the areas pointed out for clarifications below.
Comment 1: My few comments are: in the introduction, it is written than ceramic glazes were first produced roughly 10,000 years ago in the ancient Near East but they would rather appear around 1500 BC in the Near East (as mentioned in the reference)
Response 1: We were mistakenly referring to vitreous materials rather than ceramic glazes specifically. Egyptian blue faience and similar materials are much older than glazed clay bodies. We have clarified the first sentence of the introduction to the following: “Ceramic glazes, or glassy layers on top of fired clay bodies, were first produced around 1500 BCE in the ancient Near East.” We also added another sentence on page 3 to distinguish the Egyptian blue faience as follows: “For example, Egyptian faience, a self-glazing ceramic material that became popular around 2001-3000 BCE, was most commonly blue from the addition of copper perhaps in the form of corroded copper metal or copper-containing minerals like malachite or azurite.”
Comment 2: legends of Figure 1 and 2 are reversed. In the text, the references to Fig 1 and 2 have to be checked
Response 2: We would like to thank the referee for pointing this out. The reversal was caused by a formatting error when saving the document between different software/file types. We have updated the formatting so the legends match with the correct figure.

Reviewer 3 Report
According to the title of the submitted paper explores the colors of Cu-bearing pigments and their origins in pottery glazes. Perhaps the title could be more explicit as one could misleadingly think that the paper is a kind of review on the several existing Cu-bearing pigments but instead of this, is a paper basically restricted to an investigation on malachite and CuO.
The research plan is very well designed from the experimental point of view. Perhaps the set of characterization techniques used is not the ideal to target the pigments transformations intended to be tracked but some worthy results are obtained.
The results are clearly presented, although in some cases are not really linked to the unveil the origin of the observed colors. Within the discussion, some of the hypotheses are rather logical speculations not really based on the experimental results and in some cases I think that the authors could get more information on their intended target even keeping the set of techniques they use. Despite the fact that the results are not really of high scientific soundness, I consider the paper to be worth of publication after some minor changes.
Below some specific comments:
Here it is my summary of results:
-Optical properties results: they describe color and physical changes of the simples. Reflectance measurements are used to verify a shift towards bluer colors.
-SEM results are used to identify the formation of the amorphous glaze.
-XRD is used to verify the formation of the amorphous glaze from the crystalline ingredients.
-EPR is used to identify Fe3+, Cu2+ and to infer a Mn2+->Mn3+ oxidation
-FTIR is used to identify water loss.
Therefore, I find a lack of real characterization of the causes of the color changes that are merely described from the optical characterization. The evolution of the Cu pigment could have been tracked using local probe techniques, in particular microXRD (to follow the destruction of Cu crystalline phases and the eventual formation of new microcrystalline phases embedded in the glaze), local XRF and XAS to track the oxidation state changes, in particular that of Cu in the sample containing SiC. I lack the required expertise to judge the EPR results and interpretation, but I wonder why this technique has not been used to investigate the oxidation of Cu in the sample containing SiC. Also, SEM-EDX could be used to track that least the diffusion of Cu within the glaze.
One of the most interesting and challenging goals of the produced glazes would be to contribute to the Cu0 or Cu2O discussion about the origin of the red dots in the sample that contained SiC, but this is not even explored within the paper.
The decomposition of malachite is very likely to produce CuO, CO2 and H2O as suggested but this is rather an speculation than a result from the presented experimental data. Again, local probe techniques could identify if this is the transformation that really occurs.
Please find more specific comments in an attached annotated version of the submitted manuscript.

Author Response
General comments: According to the title of the submitted paper explores the colors of Cu-bearing pigments and their origins in pottery glazes. Perhaps the title could be more explicit as one could misleadingly think that the paper is a kind of review on the several existing Cu-bearing pigments but instead of this, is a paper basically restricted to an investigation on malachite and CuO.
The research plan is very well designed from the experimental point of view. Perhaps the set of characterization techniques used is not the ideal to target the pigments transformations intended to be tracked but some worthy results are obtained. The results are clearly presented, although in some cases are not really linked to the unveil the origin of the observed colors. Within the discussion, some of the hypotheses are rather logical speculations not really based on the experimental results and in some cases I think that the authors could get more information on their intended target even keeping the set of techniques they use. Despite the fact that the results are not really of high scientific soundness, I consider the paper to be worth of publication after some minor changes.
Responses: Thank you for your insightful suggestions. We have changed the title to “Exploring the colors of copper-containing pigments, copper (II) oxide and malachite, and their origins in ceramic glazes” to specify our research area. We have made all necessary changes based on your feedback. In addition, we also carefully edited the entire manuscript to make sure our ideas and results are clearly presented. We believe our revised manuscript is much stronger. Please see the following point-by-point responses.
Comment 1: I find a lack of real characterization of the causes of the color changes that are merely described from the optical characterization. The evolution of the Cu pigment could have been tracked using local probe techniques, in particular microXRD (to follow the destruction of Cu crystalline phases and the eventual formation of new microcrystalline phases embedded in the glaze), local XRF and XAS to track the oxidation state changes, in particular that of Cu in the sample containing SiC. I lack the required expertise to judge the EPR results and interpretation, but I wonder why this technique has not been used to investigate the oxidation of Cu in the sample containing SiC. Also, SEM-EDX could be used to track that least the diffusion of Cu within the glaze.
Response 1: Thank you for your valuable suggestion. We added a few paragraphs in the discussion section to address the chemical origin of the color in the glaze and how EPR is most beneficial for probing the oxidation state and the surrounding chemical environment of copper ions. Although EPR cannot detect Cu+1due to a d10 electronic configuration with no unpaired electrons, making it undetectable by EPR. However, The d9 configuration of Cu2+ means that its compounds are paramagnetic, making EPR of Cu(II) containing species a useful tool for both structural and mechanistic studies. The disappearance of the EPR signal can also tell that the sample has no Cu2+. While we could have attempted conducting EPR analysis on the CuO/SiC sample to try to see if the Cu2+ signal disappears, the reduction (red spots) was only in very localized areas, and the way we collected glaze samples would have included non-reduced areas meaning the EPR spectra of the copper(II) would still appear. Thus, local XRF and XAS to track the oxidation state changes, in particular, that of Cu in the sample containing SiC would be helpful in the future study. Since the final glaze is amorphous glassy materials, the micro-XRD may not be very useful to examine the final glaze.
Comment 2: One of the most interesting and challenging goals of the produced glazes would be to contribute to the Cu0 or Cu2O discussion about the origin of the red dots in the sample that contained SiC, but this is not even explored within the paper.
Response 2: We added a discussion on the origin of red dots that are attributed to Cu1+ or Cuo and a sentence “More careful analyses such as XANES would help to confirm the underlying chemical changes that result in these red areas” in the discussion.
We recognize that future studies using analytical techniques could help further elucidate the reduction product. For example, XANES would be useful in confirming the mechanisms of copper reduction from 2+ to 1+ or even to 0 (nanoparticles). Bubbles in our CuO/SiC glaze suggested that there was more CO2 production in this glaze than in the other glazes. We look forward to future analyses that can be used to further investigate the mechanisms of the glaze color development we have observed in our study.
Comment 3: The decomposition of malachite is very likely to produce CuO, CO2 and H2O as suggested but this is rather an speculation than a result from the presented experimental data. Again, local probe techniques could identify if this is the transformation that really occurs.
Response 3: Thank you for your insightful comments. We agree that local probe techniques could confirm this transformation, however, previous studies, such as Frost et al. (2002), have already experimentally confirmed the decomposition of natural malachite into CuO, CO2, and H2O.
